# Thermo-Mechanical and Fungi Treatment as an Alternative Lignin Degradation Method for *Bambusa oldhamii* and *Guadua angustifolia* Fibers

**DOI:** 10.3390/jof8040399

**Published:** 2022-04-14

**Authors:** Luis Garzón, Jorge I. Fajardo, Román Rodriguez-Maecker, Ernesto Delgado Fernández, Darío Cruz

**Affiliations:** 1Grupo de Investigaciónen Nuevos Materiales y Procesos de Transformación (GiMaT), Universidad Politécnica Salesiana, Calle Vieja 12-30 y Elia Liut, Cuenca 010102, Ecuador; jfajardo@ups.edu.ec; 2Carrera de Ingeniería Petroquímica, Universidad de las Fuerzas Armadas–ESPE, Latacunga 050152, Ecuador; rnrodriguez@espe.edu.ec; 3Grupo de Investigaciónen Biotecnología Ambiental INBIAM, Universidad Politécnica Salesiana, Calle Vieja 12-30 y Elia Liut, Cuenca 010102, Ecuador; mdelgado@ups.edu.ec; 4Grupo de Investigación Ecología y Evolución de Sistemas Microbianos, Departamento de Ciéncias Biológicas y Agropecuarias, Universidad Técnica Particular de Loja, San Cayetano Alto, Calle París, Loja 110107, Ecuador; djcruz@utpl.edu.ec

**Keywords:** *Bambusa oldhamii*, principal component analysis, FTIR spectroscopy, ITS-5.8S, lignin, *Fusarium* spp.

## Abstract

Different strategies have been used to degrade the molecular structure of lignins in natural fibers. Both chemical and biological processes can obtain different types of lignins for industrial use. In this study, a variation of the spectral intensity of the thermo-mechanical and fungi-modified *Bambusa oldhamii* (giant bamboo) and *Guadua angustifolia* Kunt fibers were examined via Fouriertransform infrared spectroscopy. The giant bamboo and *Guadua angustifolia* Kunt specimens were modified using a non-chemical alternative steam pressure method for degrading lignins, followed by mechanical sieving to obtain fibers of different lengths. The obtained fibers were treated with the *Fusarium incarnatum-equiseti* MF18MH45591 strain in a 21 d degradation process. The samples were subjected to Fouriertransform infrared spectroscopy before and after the strain treatment. The intensity variation was found to be in the spectral range of 1200 cm^−1^ to 1800 cm^−1^, in which lignin components are commonly found in most plant species. A multivariate analysis of the principal components of the treated and untreated control samples confirmed the changes in the spectral region of interest, which were associated with the thermo-mechanical and fungal treatment.

## 1. Introduction

In botanical terms, bamboo is a monocotyledon and several studies have been made on its microstructural and nano-characterization [1,2,3,4]. *G. angustifolia* belongs to the *Poaceae* grass family and is the most important type of native bamboo in South America, with approximately 30 species distributed from Mexico to Argentina [5]. Bamboo fiber is often brittle when compared with other natural fibers because they have a high percentage of lignins (approximately32%) [6]. The chemical composition of bamboo or *G. angustifolia* is similar to that of wood. Wood and other lignocellulosic materials are formed by three primary polymeric constituents, i.e., cellulose, lignins, and hemicelluloses.

Cellulose, considered to be the most plentiful biopolymer in nature, confers strength and stability to the plant cell walls. It has been intensively studied morphologically, mechanically, and thermally in natural fibers in recent years [7,8,9]. Lignins are a three-dimensional network built up of dimethoxylated (syringyl, S), monomethoxylated (guaiacyl, G), and non-methoxylated (p-hydroxyphenyl, H) phenylpropanoid units, derived from the corresponding p-hydroxycinnamyl alcohols, which give rise to a variety of subunits including different ether and C–C bonds [10]. In addition, lignins are highly resistant towards chemical and biological degradation, and confers mechanical resistance to wood. Compared with wood, however, the durability of bamboo against fungal and insect attack is strongly associated with its chemical composition [11,12,13].

Cellulose and hemicellulose have abundant potential as feedstock for the production of biofuels, chemicals, polymeric reinforcement, biofibres, and biopulp. However, both cellulose and hemicellulose are infiltrated by lignins [14,15]. Different pretreatments to breakdown the lignin barrier have been reported, i.e., chemical, physical, and biological methods. Physical and chemical pretreatments, i.e., mercerization and acid steam, that can obtain micro fibrils (by breaking down molecular bonds), bleach fibers (by removing chlorophyll), and enhance their adhesive properties in a polymeric matrix are widely employed [12,16,17,18]. These pretreatments increase the accessible surface area and decrease the lignin content and cellulose crystallinity as well as the degree of polymerization [19].

Biological methods have positioned themselves as an eco-friendly alternative due to their advantages over other pretreatments. Most Basidiomycetes, i.e., *Phanerochaete chrysosporium*, which produce whiterot have been used for these purposes. Furthermore, microfungi that can break down the soluble products of lignin transformation during the production of some materials have been poorly applied [20].

All natural fibers have a vibrational infrared fingerprint that identifies the different compounds, i.e., cellulose, lignins, and hemicellulose bands, as shown in Table 1. The chemical structure of lignocellulosic compounds, i.e., the aromatic ring, produces strong absorbance in the ultraviolet (UV) region associated with the molecular electronic transitions. This process provides a unique spectrum in addition to the absorbance in the infrared (IR) region, due to vibrational molecular motions.

The aim of this work was to propose an eco-friendly alternative method of degrading lignins in *G. angustifolia* and *B. oldhamii* samples via thermo-mechanical and fungi methods, as follows and depicted in Figure 1:(1)Collect the sample (natural fibers and agro industrial waste), in this particular case, *B. oldhamii* Munro the Giant Bamboo (GB) and *G. angustifolia* Kunt (GAK);(2)The thermo-mechanical process, i.e., using steam pressure and mechanical sieving to obtain microfibers of different lengths according to American Society for Testing and Materials, ASTM standard E11 (2017);(3)The fungal treatment, i.e., the microfibers of GB and GAK were then treated with *F. incarnatum-equiseti* MF18MH45591 strain in a 21 d degradation process;(4)The untreated control and treated samples were analyzed via optical microscopy and Fourier transform infrared (FTIR) spectroscopy; and(5)A multivariate analysis of the principal components was employed to confirm possible changes in the spectral region of interest (ROI).

## 2. Materials and Methods

### 2.1. Materials

*Bambusa oldhamii* Munro, i.e., giant bamboo (GB), and *Guadua angustifolia* Kunt (GAK) fibers were isolated from commercially available bamboo grown near the Ecuadorian coast, where the environmental conditions are as follows: 320 m above sea level, an average temperature of 24 °C, and a relative humidity of 70%. The culm fibers were isolated via steam explosion (ELECON-ESE-01, Elecon Electro-Constructora, Cuenca, Ecuador) at 200 psi, which is an eco-friendly method that does not require chemical reagents or energy. Steam explosion pretreatment has been shown to be very effective when applied to plant biomass [27]. In general terms, the pretreatment consists of placing vegetal material and water at a specific ratio inside a reactor Elecon-ESE-01 (Automatic or manual, Elecon Electro-Constructora, Cuenca, Ecuador), then the temperature and pressure are raised to produce steam at a high temperature; after a set reaction time, the pressure is suddenly reduced. The process steps include self-hydrolysis, penetration of high-pressure vapor into the bamboo cell walls, and explosion. During the self-hydrolysis stage, the hemicelluloses are dissolved and partially degraded, the lignins are melted and undergo a coalescence process. During the explosion stage, the plant material is divided into small fiber bundles. This process increases the surface area of carbohydrates available for enzymes, which facilitates the digestibility of the plant biomass, especially when combined with other subsequent treatments. The result of the treatment is the substantial decomposition of the lignocellulosic structure, the hydrolysis of the hemicellulose, the depolymerization of the lignin components, and consequently the separation of the cellulose fibers. The severity factor describes the effect of the combination of both the temperature and reaction time, as shown in Equation (1),
(1)$$severity=log\left(\int_{0}^{t}exp\left(T_{exp}-\left(\frac{100}{14.75}\right)\right)dt\right)$$
where *T_exp_* is the hydrolysis temperature (°C) and *t* is the reaction time (min). The vegetal chips of the two bamboo species were selected with segment lengths between 1 cm to 8 cm. Subsequently, isolation via steam explosion was carried out, and the severity factor was kept at 3.3. Additionally, after isolation, the bamboo fibers were dried to obtain a moisture content between 2 and 4. Due to the aforementioned reasons, steam explosion pretreatment is an economical process, that has a low environmental impact, avoids the use of chemical reagents, and its energy consumption is lower compared to the mechanical insulation process. Steam explosion is a process in which vegetal chips are treated with hot steam (180 °C to 240 °C) under pressure (1 MPa to 3.5 MPa), followed by an explosive decompression of the biomass that results in a rupture of the rigid structure of the biomass fibers.

Afterwards, the bamboo bundles were ground and sieved according to ASTM standard E11 [28] using a Retsch cutting mill SM 100 mill and an sieve shaker AS 200 control (Retsch, Haan, Germany). Three different fiber sizes were obtained, i.e., 150 µm, 250 µm, and 425 µm, as shown in Figure 2. The fibers were dried at a temperature of 80 °C overnight to stabilize the moisture content to less than 2%.

### 2.2. Fungal Isolation and Morpho-Molecular Characterization

The fungal strains used in the evaluation of the fiber delignification process were isolated from giant bamboo and *G. angustifolia* Kunt. The macroscopic analysis of fungal isolates was based on certain characteristics, i.e., color and mycelial growth pattern, viathe solid potato dextrose agar (PDA) cultivation method, which was incubated at a temperature of 25 °C. Different development structures were sought in the microscopic characteristics, i.e., the shape and size of the conidia and phialides as well as the type of conidiogenesis, in addition to the shape and diameter of the hyphae.

The different preparations were stained with a 1% Phloxine and decolorized with 10% KOH. The fibers were examined under a 100× magnification optical microscope. The measurements (30 on total) for each fungi structure including spores were registered. The measure ranges for the species were calculated according to the mean values and standard deviation (SD). The different morpho-species were identified according to the keys given by Seifert and Gams [29].

The total DNA extraction of the different specimens was carried outfollowing the Pure Link TM Plant Total DNA Invitrogen extraction protocol on a small amount of mycelium, while avoiding agar as much as possible. The obtained DNA was amplified via PCR using the universal primers: ITS1 (5′-TCCGTAGGTGAACCTGCGG-3′ and NL4 (5′-GGTCCGTGTTTCAAGACGG-3′), therefore obtaining the partial ITS1-5.8S-LSU region (D1/D2) [30]. The PCR conditions were as follows: 35 cycles, starting with an initial denaturation at a temperature of 94 °C for 3 min.

Each cycle involved a denaturation step at a temperature of 94 °C for 30 s. The annealing temperature was 55 °C for 30 s, followed by an extension at 72 °C for 2 min, and a final extension at 72 °C for 10 min. The total PCR reaction volume was 20 µL, with 18 µL of Invitrogen Platinum TM PCR Supermix, 0.2 µL of each primer, 0.4 µL of 10% BSA (bovine serum albumin), and 1.4µL of DNA. The PCR products were verified via electrophoresis in 1% agarose gels plus 1X Red Gel solution (Biotium); the running conditions were 128 V and 300 mA for 25 min. In addition, 1.5µL of 1 Kb DNA ladder (Invitrogen, Waltham, MA, USA) was used, as well as 2µL of the PCR product plus 2µL of bromophenol blue standardized [31]. The running buffer was 1X TBE (Trisborate, EDTA) ethylenediaminetetraacetic acid. The PCR-positive products were purified using the PureLink PCR Purification Kit (Invitrogen) and sequenced by Macrogen (Seoul, Korea).

A morphomolecular analysis for the taxonomic location of the 3 strains, i.e., MF18 MH455291, and MF32-MH455293 (*F. oxysporum*), was performed. In a preliminary analysis, the three fungal isolates were assessed, showing that the delignification capacity was similar. Based on the above analysis, the isolate *F. incarnatum equiseti* MF18MH45591 was used in this study.

### 2.3. Fungal Delignification of the Vegetable Fibers 

The fibers were sterilized in an autoclave at a temperature of 120 °C for 15 min and 15 psi in 9 cm Petri dishes with PDA. Sterile cellophane paper was placed over the surface. Vegetable fiber samples weighing 2 g each were placed on the cellophane and the inoculum fungal isolate was put on the fibers. Each treatment was run in triplicate for a 21 d period.

### 2.4. Attenuated Total Reflection Fourier Transform Infrared (ATR-FTIR) Spectra

Fourier transform infrared spectroscopy (FTIR) measurements were made on a Frontier FTIR spectrophotometer (Perkin Elmer, Waltham, MA, USA), coupled to a universal attenuated total reflectance (ATR) sampling accessory equipped with a zinc selenide measurement cell (Perkin Elmer, Waltham, MA, USA). All fiber samples (untreated and biologically treated) were dried for 48 h in a vacuum oven at a temperature of 50 °C, and then stored in a vacuum desiccator at room temperature. Then, the samples were placed directly on the ATR sampling accessory window crystal and scanned at a range of 4000 cm^−1^ to 650 cm^−1^, with a resolution of 8 cm^−1^, and an average of 5 accumulations.

### 2.5. Principal Component Analysis

Principal component analysis (PCA) is a multivariate technique that analyzes a set of data in which observations are described by several dependent variables. In this case, the interest of the authors was to detect changes in the FTIR signal in the specimens treated with fungi extracted from their own fibers. Principal component analysis uses a vector space transform to reduce the dimensionality of large data sets. The original data set, which may involve many variables, can be reduced to just a few variables. Important information can be extracted from a new coordinate system, called the principal components, obtained from a rotation of the original data set [32].

For the purpose of the PCA, the variables were described as giant bamboo (GB) and *G. angustifolia* Kunt (GAK) treated for 21 d with the *F. incarnatum-equiseti* strain and the untreated samples (control). In each case, the samples were obtained through a mechanical process via stem explosion and then by selecting lengths of 150 µm, 250 µm, and 425 µm. The ATR-FTIR spectral record of each sample was placed in an *n* × *m* matrix, where the spectral region of interest (ROI) was from 1200 cm^−1^ to 1800 cm^−1^. The aim of PCA is to linearly transform this matrix, X into another matrix Z, as shown in Equation (2),
(2)$$Z=\lambda^{T}X$$
where Z is the vector of the principal components and λ^T^ is the matrix of the coefficients λij for i, j = 1, 2, …, m. The first principal component (Z1) is the linear combination of the original features, as shown in Equation (3),
Z1 = λ_11_ X_1_ + λ_12_ X_2_ +…+ λ_1m_X_m_(3)
where the first component is chosen to have the largest possible variance of them features. The second principal component (Z2) is chosen to have the second largest variance of X1, Xm while being uncorrelated with Z1, and so on for the remaining principal components. The KMO (Kaiser–Meyer–Olkin) index test was initially carried out to check whether the original variables could be efficiently factorized. Bartlett’s test of sphericity was used to check any redundancy between the variables that could be summarized in a smaller number of factors. If the KMO index is high (approximately 1), the factorial and PCA can be employed and if the KMO is low (0), then the PCA will not be relevant. Statistical analysis was performed using SPSS software (SPSS, version 22, Chicago, IL, USA).

## 3. Results

### 3.1. Morphological Characteristics

The morphological macro- and microscopic characteristics (as shown in Figure 3 and Figure 4) of the strains allowed for the determination of the species was in the genus *Fusarium*, which has white-creamy or pinkish-purple aerial mycelium, which resembles either *F. oxysporum* (as shown in Figure 3a) or *F. equiseti* (as shown in Figure 3b). Acquisition of images over30 days. 

### 3.2. Phylogenetic Analysis

The morpho-species were corroborated by phylogeny, as shown in Figure 5, which clearly distinguished three genotypes belonging to species of the genus *Fusarium*. The specimen MF32-MH455293 corresponds to *F. oxysporum* with a 98% bootstrap, while the two remaining sequences were from MF18MH455291 and MF52 MH455292 strains, which fell into the clade *F. incarnatum-equiseti* species complex (FIESC), maintaining a variation between the species with less than a 1% genetic distance.

### 3.3. Fungal Delignification of the Vegetal Fibers

*Fusarium incarnatum-equiseti* strain (MF18) with sequence accession number (MH455291) was the best lignin fiber decomposer according to the comparison of the microscopic characteristics.

### 3.4. Attenuated Total Reflection Fourier Transform Infrared (ATR-FTIR)

Figure 6 shows the ATR-FTIR spectra for the untreated (Figure 6a,c) and the treated (Figure 6b,d) GB and GAK natural fibers by lengths. Differences in the peak intensities between the wave numbers were found by comparing the ATR-FTIR spectra for giant bamboo (GB) and *G. angustifolia* Kunt (GAK) with fiber lengths of 150 µm, 250 µm, and 425 µm, for both the untreated (as shown in Figure 6a,c) and treated (as shown in Figure 6b,d). In both cases, the ATR-FTIR spectra revealed considerable changes within the spectrum region of interest, which was between 1200 cm^−1^ to 1800 cm^−1^. Peaks isolated in the region from 2800 cm^−1^to 3500 cm^−1^, where stretching vibrations in the methyl and methylene groups and all types of hydroxyls involved in forming intramolecular H-bonds are found, were also affected by the treatment.

Figure 7 shows the regions of spectra in which collections of characteristic lignin peaks for GB and GAK samples are found. Changes in the intensities of the polymer functional groups were analyzed from each labeled peak as well as those that were reduced or disappeared due to treatment, within the range of interest 1200 cm^−1^ to 1800 cm^−1^. All spectra were placed at the same intensity reference base for comparison.

### 3.5. Principal Component Analysis

Of the ATR-FTIR results obtained for the untreated controls (GB and GAK) and the treated samples, a data matrix was constructed of the values from the spectral range intensities from 1200 cm^−1^ to 1800 cm^−1^. From the obtained data matrix, the spectral band matrix scores for GB and GAK are represented as follows:

Case 1: GB

X1 = Giant Bamboo 150 µm treated;

X2 = Giant Bamboo 250 µm treated;

X3 = Giant Bamboo 425 µm treated;

X4 = Giant Bamboo 150 µm untreated control;

X5 = Giant Bamboo 250 µm untreated control;

X6 = Giant Bamboo 425 µm untreated control.

Case 2: GAK

In the case of untreated and treated *G. angustifolia*Kunt (GAK):

X7 = GAK 150 µm treated;

X8 = GAK 250 µm treated;

X9 = GAK 425 µm treated;

X10 = GAK 150 µm untreated control;

X11 = GAK 250 µm untreated control;

X12 = GAK 425 µm untreated.

A PCA biplot, as shown in Figure 8, shows the loadings of variables for the GB and GAK cases. As shown, variables X1, X2, X3, X7, X8, and X9 have the most influence on the second component. Interestingly, these variables are associated with the samples treated via thermo-mechanical and fungal methods. Variables X4, X5, X6, X10, X11, and X12 lie on the first component and were associated with the untreated control samples.

## 4. Discussion

The results of this study on giant bamboo (GB) and *G. angustifolia* Kunt (GAK) with different length samples and treated with the *F. incarnatum-equiseti* strain MF18MH45591 for 21 d showed noticeable variations in the intensities of the ATR-FTIR spectra in the region of interest for lignin components. Many of the characteristic lignocellulose bands overlap with cellulose and hemicelluloses at the range of1369 cm^−1^ to 1883 cm^−1^. The peaks at 1464 cm^−1^ to 1530 cm^−1^ for lignin components, which were in the ROI, had approximately 50% less intensity compared to the untreated samples for both the GB and GAK samples. Similar results have been reported for the *Dipterocarpaceae* wood species [33]. Thus, it can be inferred that the biodegrading activity is due to the production of primary and secondary metabolites (enzymes) that probably cause the multiple reactions which determine the percentage of fiber delignification.

As shown in Figure 6, on the GB 250 μm curve of the treated samples, the individual peaks at 1397 cm^−1^, 1438 cm^−1^, and 1540 cm^−1^ were reduced in their intensities, whereas the GAK425 μm curve of the treated samples had peaks that disappeared after treatment. A closer inspection is shown in Figure 7. The multivariate technique identified the major effects of both the fiber length and the treatment applied in the X1, X2, and X3 (GB150 μm, GB250 µm, and GB425 µm) and X7, X8, and X9 for GAK samples on the changes in the characteristic peaks of the ATR-FTIR spectra, also as shown in Figure 8.

There are several extant studies on the pre-biological treatment of vegetable fibers with wood white-rot fungi; these fungal isolates can break down and mineralize recalcitrant ligninsthrough the production of enzymes, i.e., laccase, lignin peroxidase (LiP), manganese peroxidase (MnP), aryl-alcohol oxidase (AAO), and polyphenol peroxidase (PPO) [29,34,35,36]. The group of lignocellulosic fungi included species of *Ascomycetes*, i.e., *Aspergillus*, *Penicillium*, and *Trichoderma reesei*, in addition to certain *basidiomycetes* [37,38]. However, many *Fusarium* species have been reported to promotelignin degradation due to the extracellular enzymatic activity through hydrolase production, which degrades polysaccharides and laccases [36,37]. In addition, *Fusarium* species cause oxidative degradation with various substances, i.e., aryl-alcohol oxidase, which causes decomposition, and the exchange of organic matter in natural ecosystems [38].

New studies should be carried out to understand if *F. incarnatum-equiseti* can be utilized as a new alternative non-chemical pretreatment for fibers rich in lignins in order to replace environmentally harmful chemical treatments.

## 5. Conclusions

The culm fibers of GAK and GB were isolated via steam explosion (at 200psi), with the severity factor controlled at 3.3. Mechanical sieving, according to ASTM standard E11 (2017), was performed to obtain microfibers of different lengths (150μm, 250μm, and 425μm).

The fungal strains used in the evaluation of the fiber delignification process were isolated from *Bambusa oldhamii* and *G. angustifolia* Kunt microfibers after thermo-mechanical treatment. With the severity factor controlled at 3.3 via steam explosion, it allowed a defibrillation of the samples without extreme damage and their subsequent mechanical screening.

The GAK and GB microfibers were treated with the *F. incarnatum-equiseti* strain, with the accession number MF18MH45591, in a 21 d degradation process, minimum treatment time where functional changes were detected by FTIR-ATR and multivariate technique. The multivariate analysis of the principal components was employed to confirm any noticeable changes in the spectral region of interest (1200 cm^−1^ to 1800 cm^−1^), where individual peaks at 1484 cm^−1^, 1505 cm^−1^, 1540 cm^−1^, 1600 cm^−1^, and 1615 cm^−1^were found to be approximately 50% less intense than the untreated control samples. Other intensity peaks in the GAK samples simply disappeared. Future works should be aimed at increasing the severity factor to reduce the time treatment with genus *Fusarium*.

## Figures and Tables

**Figure 1 jof-08-00399-f001:**
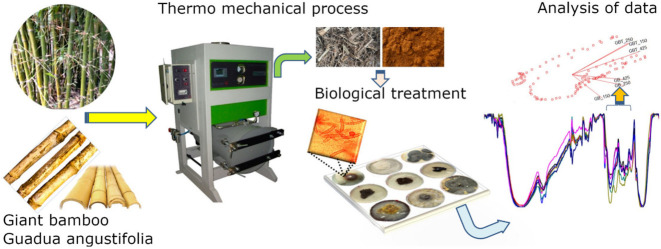
Eco-friendly proposal to modify the chemical structure of lignocellulosic compounds and break down the lignin linkages in natural fibers.

**Figure 2 jof-08-00399-f002:**
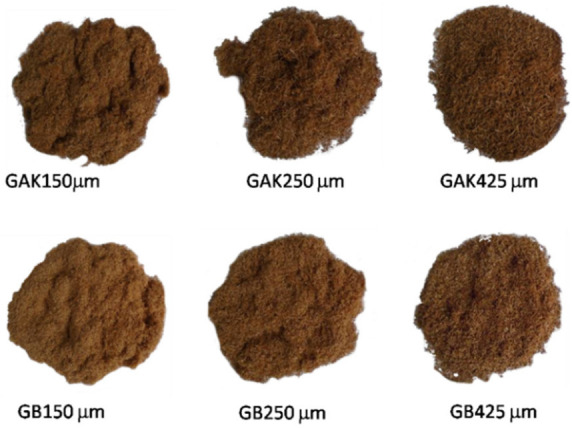
Fiber sizes obtained from the pretreated GAK and GB, according to ASTM standard E11-17.

**Figure 3 jof-08-00399-f003:**
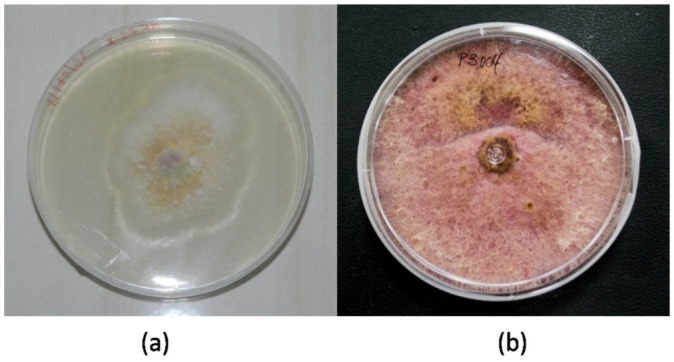
*Fusarium* specimens: (**a**) *F. oxysporum* MF32-MH455293 and (**b**) *F. equiseti* MF18-MH455291.

**Figure 4 jof-08-00399-f004:**
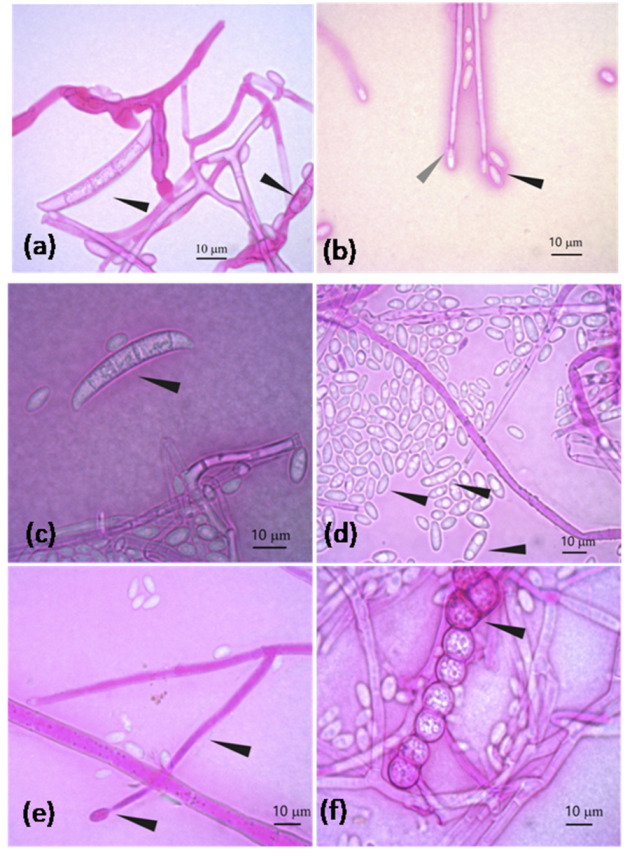
*F. oxysporum* characteristics: (**a**) Banana-shaped macroconidia transversely septated and macroconidia in formation; (**b**) Oval and elongated microconidia (black arrowhead) and elongated conidiophore in conidiogenesis at the end (gray arrowhead) (note: terminal clamidospores were not observed in (**b**)). *F. equiseti* characteristics: (**c**) Banana-shaped macroconidia and transversely septated; (**d**) Oval and elongated microconidia; (**e**) Elongated mono-phialide with a transverse septum and conidiogenesis at the end; (**f**) Chain of verrucous clamidospores.

**Figure 5 jof-08-00399-f005:**
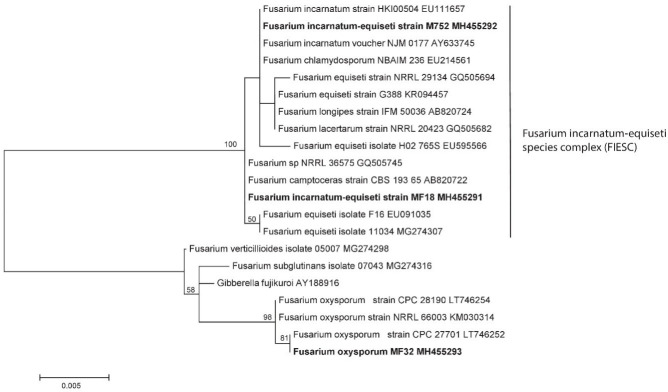
Genotypes belonging to species of the genus *Fusarium* obtained from samples. The mid-point rooting phylogenetic tree concatenated with maximum likelihood (1000 of bootstrap) for regions ITS-5.8S and partial LSU D1/D2 (Note: Bootstrap values above 50 are shown on the nodes).

**Figure 6 jof-08-00399-f006:**
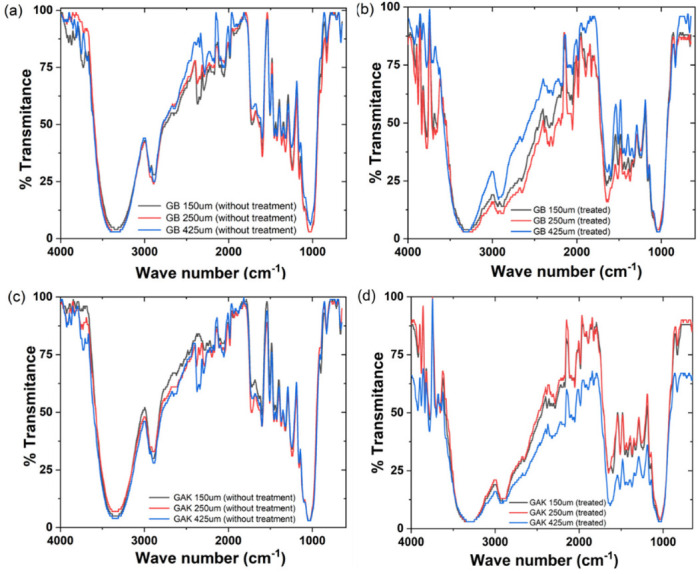
ATR-FTIR results of the giant bamboo (GB) and *G. angustifolia* Kunt (GAK) fibers at different sieves: (**a**,**c**) the untreated control and (**b**,**d**) with treatment, respectively.

**Figure 7 jof-08-00399-f007:**
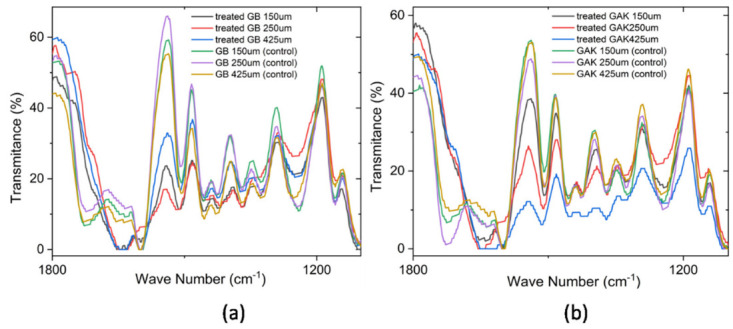
Comparison of the ATR-FTIR spectra at a range of 1800 cm^−1^ to 1200 cm^−1^ for the (**a**) treated GB and control and (**b**) treated GAK and control samples. A correction baseline was performed.

**Figure 8 jof-08-00399-f008:**
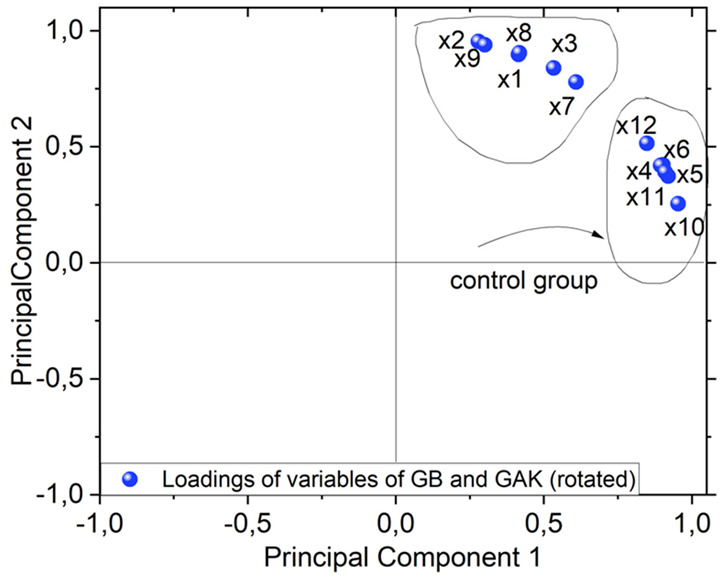
PCA biplot of both sample giant bamboo GB and *G. angustifolia* Kunt GAK showing clustering (untreated or control group pointed out with black arrow and treated samples).

**Table 1 jof-08-00399-t001:** Vibrational characteristic for natural fibers obtained via FTIR.

Natural Fiber	Wave Number (cm^−1^)	Description	References
Jute/Ramie	3400	OH stretching	[21,22,23]
	2900	CH stretching	
	1040 to 1060	CO stretching vibration cellulose	
	1550–1650	Aromatic skeleton lignin	
Bamboo	3800 to 3300	OH stretching vibration	[24]
	2921.63	CH stretching cell/hemicellulose	
	2853.17	OH stretching vibration of inter and intramolecular H-bonding	
	1733.69	CO stretching of carboxylic acid or ester	
	1636.3	Absorbed water	
	1608.34	C=C stretching vibration of lignin	
	1464 to 1530	Lignin components	
	1436.7	>CH2 bending in lignin	
	1419.3	>CH2 and =CH3 bending	
	1384.6	CH bending	
	1363	>CO stretching of acetyl ring	
	1339	>CO stretching of acetyl ring	
	1160.9	Antisymmetric bridge C=0=C stretching	
Coconut	3340	OH hydrogen bonded	[25]
	1728	Bands of hemicellulose	
	1390	Antisymmetrical deformation of CH in cellulose and hemicellulose group	
	1370	Symmetrical deformation of CH in cellulose and hemicellulose group	
	1238	CO vibration of esters, ethers, and phenols	
Arundo Donax	3400	OH stretching vibration hydrogen bond of hydroxyl group	[26]
	2923	CH stretching vibration from CH and CH2 in cellulose and hemicellulose	
	1730	C=O stretching vibration of the acetyl group in hemicellulose	
	1594	Water in fibers	
	1506	C=C stretching of benzene ring in lignin	
	1422	CH2 symmetric bending	

## Data Availability

Not applicable.

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
