# Peer review of "Thermo-Mechanical and Fungi Treatment as an Alternative Lignin Degradation Method for Bambusa oldhamii and Guadua angustifolia Fibers"

_jof, 2022, doi:10.3390/jof8040399_

Round 1

Reviewer 1 Report

Dear authors.
The paper has been carefully reviewed. Some considerations are made below.
- Why insert Figure 1 in the introduction? Is it really necessary? It looks like a graphical abstract.
- In the results, figures 7 and 8 are of poor quality. It looks like the peaks are clipped at zero. This is normal?
Congratulations on the search. Interesting contribution.

Author Response

I am very sorry for the late response to your comments. Unfortunately, due to
health situations, I was not able to send it in the established time.

Author Response

Dear reviewer,
I am very sorry for the late response to your comments. Unfortunately, due to
health situations, I was not able to send it in the established time.

Authors accept the observations indicated in the PDF document.
I send the corrections in word as new document.

Round 2

Reviewer 2 Report

Author implemented all the suggested corrections.